# LEARNING DISCRIMINATORS AS ENERGY NETWORKS IN ADVERSARIAL LEARNING

## ABSTRACT

We propose a novel adversarial learning framework in this work. Existing adversarial learning methods involve two separate networks, i.e., the structured prediction models and the discriminative models, in the training. The information captured by discriminative models complements that in the structured prediction models, but few existing researches have studied on utilizing such information to improve structured prediction models at the inference stage. In this work, we propose to refine the predictions of structured prediction models by effectively integrating discriminative models into the prediction. Discriminative models are treated as energy-based models. Similar to the adversarial learning, discriminative models are trained to estimate scores which measure the quality of predicted outputs, while structured prediction models are trained to predict contrastive outputs with maximal energy scores. In this way, the gradient vanishing problem is ameliorated, and thus we are able to perform inference by following the ascent gradient directions of discriminative models to refine structured prediction models. The proposed method is able to handle a range of tasks, *e.g.*, multi-label classification and image segmentation. Empirical results on these two tasks validate the effectiveness of our learning method.

## 1 INTRODUCTION

This work focuses on applying adversarial learning (Goodfellow et al., 2014a) to solve structured prediction tasks, *e.g.*, multi-label classification and image segmentation. Adversarial learning can be formalized as a minimax two-player game between structured prediction models and discriminative models. Discriminative models are learned to distinguish between the outputs predicted by the structured prediction models and the training data, while structured prediction models are learned to predict outputs to fool discriminative models. Though structured prediction models are trained by the gradients of discriminative models, existing methods rarely use discriminative models to improve structured prediction models at the inference stage.

A straightforward way of utilizing discriminative models for inference is to follow the ascent gradient directions of discriminative models to refine the predicted outputs. However, due to the well-known gradient vanishing problems, the gradients of discriminative models are almost zero for all the predicted outputs. It is difficult to resolve the gradient vanishing problems, because they are caused by the training instability of the existing adversarial learning framework. Consequently, most existing methods do not use the information from discriminative models to refine structured prediction models.

Most existing adversarial learning methods take discriminative models as classifiers. If discriminative models well separate real and predicted samples, they tend to assign the same scores to all predicted samples. Differently, energy-based models (LeCun et al., 2006; Gygli et al., 2017) usually predict different energy scores for different samples. Therefore, we propose to train discriminative models as energy-based models to ameliorate the gradient vanishing problems. In our framework, discriminative models are learned to assign scores to evaluate the quality of predicted outputs. Structured prediction models are learned to predict outputs that are judged to have maximum scores by discriminative models. In this way, discriminative models are trained to approximate continues value functions which evaluate the quality of predicted output. The gradients of discriminative models are not zero for predicted outputs. Thus, we can refine structured prediction models by following the

ascent gradient directions of discriminative models at the inference stage. In this paper, we refer our method as learning discriminative models to refine structured prediction models (LDRSP).

The proposed method learns discriminative models utilizing the data generated by the structured prediction models. Gygli et al. (2017) found that the key to learning deep value networks is generating proper training data. We propose to augment the training set of discriminative models by following the data generation methods proposed in (Gygli et al., 2017). At the training stage, we simultaneously run the inference algorithm to generate extra training samples utilizing models in previous iterations. These samples are useful since they are generated along the gradient-based inference trajectory utilized at the inference stage. We also augment the training set with adversarial samples (Goodfellow et al., 2014b). These samples are used as negative samples to train the discriminative models.

To validate our method, experiments are conducted on multi-label classification, binary image segmentation, and 3-class face segmentation tasks, and experimental results indicate that our method can learn discriminative models to effectively refine structured prediction models.

This work has two contributions: (1) We propose a novel adversarial learning framework for structured prediction, in which the information captured by discriminative models can be used to improve structured prediction models at the inference stage. (2) We propose to learn discriminative models to approximate continues value functions that evaluate the quality of the predicted outputs, and thus ameliorate the gradient vanishing problems.

## 2 RELATED WORK

Recently, adversarial learning has been well studied on producing high-resolution and high-quality images (Denton et al., 2015; Karras et al., 2017), improving the training stability and getting rid of problems like model collapse (Arjovsky et al., 2017; Gulrajani et al., 2017; Mao et al., 2017; Qi, 2017), image to image translation (Isola et al., 2017; Zhu et al., 2017; Huang et al., 2018), semantic image segmentation (Luc et al., 2016; Hwang et al., 2018) and neural dialogue generation (Li et al., 2017). These methods learn discriminative models and generative models in an adversarial way, but discriminative models are abandoned once models are trained. Since discriminative models are learned to capture the discriminative distributions between the generated samples and the training data, we argue that they are able to be utilized to make the generated samples more real. As we know, we are the first to learn discriminative models that can be directly utilized to refine structured prediction models.

There are some researches that utilize iterative inference methods. Since Denoising Autoencoders (DAEs) can estimate the gradients of energy distribution functions, Nguyen et al. (2016) propose to learn DAEs to capture probability distributions of realistic images. In the inference process, DAEs can be utilized to iteratively refine generated images. Romero et al. (2017) expand similar ideas to the image segmentation. Different from these methods, in which feed-forward networks are utilized to iteratively refine the generated samples, we adopt a gradient-based inference to find samples with the zero task loss.

Learning discriminative models as energy-based models has been recently studied (Zhao et al., 2017; Dai et al., 2017). These works only constrain that discriminative models predict lower energy scores for the training data than the generated samples. However, our method learns discriminative models to approximate value functions that evaluate the quality of the generated samples. Moreover, we propose to generate extra training samples to learn discriminative models, so that discriminative models can better capture the probability distributions of the sample space.

There is a rising interest in energy-based structured prediction (Zheng et al., 2015; Chen et al., 2015; Song et al., 2016). Amos et al. (2016) proposed to add constraints to the neural network parameters such that the output of the neural network is a convex function of (some of) the inputs. Belanger & McCallum (2016) introduced Structured Prediction Energy Network (SPEN). SPEN relies on a max-margin surrogate objective to ensure that the neural network predicts the lowest energy value for the ground-truth label. Belanger et al. (2017) improved SPEN by proposing an end-to-end version of SPEN, which directly back-propagates through a computation graph that unrolls gradient-based energy minimization. Inspired by the reinforcement learning, Gygli et al. (2017) proposed a Deep Value Network (DVN) that directly learns to evaluate the quality of different output configures.

Compared with these methods, our method adversarially learns energy-based models and structured prediction models rather than learns energy-based models alone.

## 3 APPROACH

We propose a novel adversarial learning framework for structured prediction, in which discriminative models $D(\mathbf{x}, \mathbf{y}; \theta_d)$ can be used to refine structured prediction models $G(\mathbf{x}; \theta_g)$ at the inference stage.

In our method, discriminators are treated as energy-based models, which take both input objects $\mathbf{x}$ and possible outputs $\mathbf{y}$ as inputs and predict scores in the range of $[0, 1]$. We assume that at the training stage, one can get access to an oracle value function $v^*(\mathbf{y}, \mathbf{y}^*)$, which evaluates the quality of $\mathbf{y}$ corresponding to $\mathbf{x}$. Here $\mathbf{y}^*$ is the ground-truth label. The discriminators are learned to mimic the behavior of the oracle value functions. Following (Gygli et al., 2017), we utilize intersection over union (IOU) and $F_1$ metrics as the oracle value functions for image segmentation and multi-label classification, respectively, which are defined on $(\mathbf{y}, \mathbf{y}^*) \in \{1, 0\}^M \times \{0, 1\}^M$,

$$v^*_{IOU}(\mathbf{y}, \mathbf{y}^*) = \frac{\mathbf{y} \cap \mathbf{y}^*}{\mathbf{y} \cup \mathbf{y}^*}, \tag{1}$$

$$v^*_{F_1}(\mathbf{y}, \mathbf{y}^*) = \frac{2(\mathbf{y} \cap \mathbf{y}^*)}{(\mathbf{y} \cap \mathbf{y}^*) + (\mathbf{y} \cup \mathbf{y}^*)}. \tag{2}$$

Here $\mathbf{y} \cap \mathbf{y}^*$ denotes the number of dimension $i$ where both $y_i$ and $y_i^*$ are active and $\mathbf{y} \cup \mathbf{y}^*$ denotes the number of dimensions where at least one of $y_i$ and $y_i^*$ is active. $y_i$ and $y_i^*$ denote the $i$-th variable of $\mathbf{y}$ and $\mathbf{y}^*$. To apply $v^*(\mathbf{y}, \mathbf{y}^*)$ to the continuous output $\mathbf{y}$, the notions of intersection and union are extended by using element-wise min and max operators,

$$\mathbf{y} \cap \mathbf{y}^* = \sum_{i=1}^{M} \min(y_i, y_i^*), \tag{3}$$

$$\mathbf{y} \cup \mathbf{y}^* = \sum_{i=1}^{M} \max(y_i, y_i^*). \tag{4}$$

We propose to learn discriminators $D(\mathbf{x}, \mathbf{y}; \theta_d)$ to estimate $v^*(\mathbf{y}, \mathbf{y}^*)$. The learning of discriminators can be understood from a regression setting with $\mathbf{z} = (\mathbf{x}, \mathbf{y})$ as inputs and $v = v^*(\mathbf{x}, \mathbf{y})$ as the target outputs. The structured prediction models are learned to predict outputs scored highly by discriminators. Discriminators and structured prediction models are respectively learned by optimizing:

$$\min_{\theta_d} \quad \mathbb{E}_{\mathbf{x} \sim p_X} \left[ \frac{1}{2}(D(\mathbf{x}, G(\mathbf{x}; \theta_g); \theta_d) - v^*_g)^2 \right] + \mathbb{E}_{\mathbf{x} \sim p_X} \left[ \frac{1}{2}(D(\mathbf{x}, \mathbf{y}^*; \theta_d) - 1)^2 \right], \tag{5}$$

$$\min_{\theta_g} \quad \mathbb{E}_{\mathbf{x} \sim p_X} \left[ L_g(G(\mathbf{x}; \theta_g), \mathbf{y}^*) + \frac{1}{2}(D(\mathbf{x}, G(\mathbf{x}; \theta_g); \theta_d) - 1)^2 \right]. \tag{6}$$

Here $v^*_g = v^*(G(\mathbf{x}; \theta_g), \mathbf{y}^*)$, and $L_g$ is a surrogate loss. Equation 5 can be understood in two aspects: (1) It's a modified version of least-square GAN (Mao et al., 2016) loss, and discriminators are learned to predict oracle values $v^*_g$ for the samples predicted by structured prediction models; (2) It learns an energy-based model using training samples that consist of ground-truth samples $(\mathbf{x}, \mathbf{y}^*, 1)$ and predicted samples $(\mathbf{x}, G(\mathbf{x}; \theta_g), v^*_g)$. The second term of Equation 6 regularizes structured prediction models such that the predicted samples tend to have higher scores.

Once models are trained, discriminators can refine structured prediction models by utilizing a gradient-based inference. The outputs predicted by structured prediction models are updated following the ascent gradient directions of discriminators that lead to the high scores:

$$\mathbf{y}^{(0)} = G(\mathbf{x}; \theta_g), \tag{7}$$

$$\mathbf{y}^{(t+1)} = \mathcal{P}_{\mathcal{Y}} \left( \mathbf{y}^{(t)} + \eta \frac{\partial}{\partial \mathbf{y}} D(\mathbf{x}, \mathbf{y}^{(t)}; \theta_d) \right). \tag{8}$$

Here, $\mathcal{P}_{\mathcal{Y}}$ denotes an operator that projects the predicted outputs back to the feasible set of solutions. In a simple case, where $\mathcal{Y} = [0,1]^M$, the $\mathcal{P}_{\mathcal{Y}}$ operator clips the predicted outputs. It is observed in experiments, $i.e.$, the learned discriminators tend to give small gradients during the gradient-based inference. One reason is that the predicted outputs of structured prediction models are already close to the ground-truth labels. In order to further improve the predicted outputs by using the gradient-based inference and overcome the small-gradient issue, we use the normalized gradient method, $i.e.$,

$$\mathbf{y}^{(t+1)} = \mathcal{P}_{\mathcal{Y}} \left( \mathbf{y}^{(t)} + \eta \frac{\left( \frac{\partial}{\partial \mathbf{y}} D(\mathbf{x}, \mathbf{y}^{(t)}; \theta_d) \right)}{\left\| \frac{\partial}{\partial \mathbf{y}} D(\mathbf{x}, \mathbf{y}^{(t)}; \theta_d) \right\|} \right). \tag{9}$$

Gygli et al. (2017) proposed to simultaneously generate training samples to learn their energy-based model, $i.e.$, Deep Value Network (DVN) in the training process. We follow their method to generate extra training samples to learn discriminators. The training samples are a set of tuples (input $\mathbf{x}$, output $\mathbf{y}$, oracle value $v^*$) represented as $T \equiv \{(\mathbf{x}^{(i)}, \mathbf{y}^{(i)}, v^{*(i)})\}_{i=1}^N$. Here $N$ is the size of the training set, and $\mathbf{x}^{(i)}$, $\mathbf{y}^{(i)}$ and $v^{*(i)}$ respectively denote the $i$-th image, $i$-th output and $i$-th oracle value in the training set. Similar to (Gygli et al., 2017), we utilize two different methods to generate training samples:

- **Inference samples.** At each training iteration, structured prediction models are first used to predict samples. Then, we take these samples as initial solutions and run a gradient-based inference to find high-valued samples of discriminators. These samples are useful since they are generated along the inference trajectory at the inference stage. These samples can be generated during training by using models in a previous iteration (Gygli et al., 2017).

- **Adversarial samples.** Maximize the loss: $\frac{1}{2}(D(\mathbf{x}, \mathbf{y}; \theta_d) - v^*)^2$ with respect to $\mathbf{y}$ using a gradient-based optimizer (Goodfellow et al., 2014b). These samples serve as negative samples to learn discriminators. Similar to the inference samples, the adversarial samples are also generated during training.

To utilize these training samples to learn discriminators, we add another loss term to the Equation 5, and the new objective function is as follows:

$$\min_{\theta_d} \quad \mathbb{E}_{\mathbf{x} \sim p_X} \left[ \frac{1}{2}(D(\mathbf{x}, G(\mathbf{x}; \theta_g); \theta_d) - v_g^*)^2 \right] + \mathbb{E}_{\mathbf{x} \sim p_X} \left[ \frac{1}{2}(D(\mathbf{x}, \mathbf{y}^*; \theta_d) - 1)^2 \right]$$
$$+ \mathbb{E}_{(\mathbf{x}, \mathbf{y}, v^*) \sim T} \left[ \frac{1}{2}(D(\mathbf{x}, \mathbf{y}; \theta_d) - v^*)^2 \right]. \tag{10}$$

We utilize the Adam optimizer (Kingma & Ba, 2014) with the momentum term $\beta_1 = 0.5$ to train structure prediction models and discriminators. At each training iteration, we randomly generate inference or adversarial samples and update the parameters of the structured prediction models and the discriminators according to Equation (6) and Equation (10). The models are trained until convergence.

## 4 EXPERIMENTS

Experiments are conducted on three tasks: multi-label classification, binary image segmentation, and 3-class face segmentation. We compare our LDRSP to other state-of-the-art adversarial learning methods on these tasks, and results are reported in Section 4.1, 4.2 and 4.3. The code is implemented using the deep learning framework, $i.e.$, Tensorflow (Abadi et al., 2016).

### 4.1 PREDICTION PERFORMANCE ON MULTI-LABEL CLASSIFICATION

We use standard benchmarks of this task, namely Bibtex and Bookmarks introduced by (Katakis et al., 2008) in this section. On these two datasets, tags need to be predicted for text inputs and multiple labels are possible for each input. A two-layer neural network (Belanger & McCallum, 2016) is utilized as our structured prediction model. The same network architecture of DVN is adopted as our

discriminator, which consists of one or two hidden layers with Softplus non-linearities. Following (Belanger & McCallum, 2016), a cross-entropy loss is used as the surrogate loss.

Besides the proposed LDRSP, different adversarial learning methods, *e.g.*, GAN (Goodfellow et al., 2014a), WGAN+GP (Gulrajani et al., 2017), LSGAN (Mao et al., 2016) and EBGAN (Zhao et al., 2017) are utilized to learn the structured prediction models and the discriminators. These structured prediction models and discriminators have the same network architecture. For all the methods, the hyper-parameter exploration is performed, and we follow the gradient directions of the discriminators to refine the structured prediction models at the inference stage.

The experimental results are reported in Table 1. For all the adversarial learning methods, we report both the prediction results of structured prediction models $G$ and the prediction results refined by the discriminators $D$. As it shows, the proposed LDRSP successfully learns $D$ that can be utilized to refine the $G$ and achieve state-of-the-art performance on both Bibtex and Bookmarks datasets. The refinement improves the performance of $G$ by 3.6 % on Bibtex dataset and 1.1 % on Bookmarks dataset. However, for other adversarial learning methods, the refinement leads to negligible improvements or even decrements in performance. For example, the refinement of EBGAN improves the performance by 0.2 % on Bookmarks dataset, and the refinement of LSGAN decreases the performance by 0.1 % on Bibtex dataset. Following (Belanger & McCallum, 2016), we implement a baseline model by training structured prediction models via minimizing a cross-entropy loss, *i.e.*, the first term of Equation (6). Comparing the performance of the baseline model and the performance of our adversarially learned structured prediction models, we notice that jointly learning structured prediction models and discriminators via the proposed method greatly improves the performance of structured prediction models.

The performance of state-of-the-art structured prediction methods *e.g.*, logistic regression, a two-layer neural network learned with a cross-entropy loss, SPEN (Belanger & McCallum, 2016), PRLR (Lin et al., 2014), and DVN (Gygli et al., 2017) is also reported in Table 1. Although our method, the DVN, and the SPEN have the same energy network architecture, our method outperforms the DVN and the SPEN on both Bibtex and Bookmarks datasets. Our method also greatly improves over feed-forward models: the logistic regression, the two-layer neural network, and the PRLP.

Table 1: The comparison of $F_1$ score between our method and other state-of-the-art methods on Bibtex and Bookmarks datasets. In the table, SPM denotes the structured prediction model.

| Method | Bibtex | Bookmarks |
|---|---|---|
| Logistic regression (Lin et al., 2014) | 37.2 | 30.7 |
| NN baseline (Belanger & McCallum, 2016) | 38.9 | 33.8 |
| SPEN (Belanger & McCallum, 2016) | 42.2 | 34.4 |
| PRLR (Lin et al., 2014) | 44.2 | 34.9 |
| DVN (Gygli et al., 2017) | 44.7 | 37.1 |
| SPM baseline | 38.9 | 32.8 |
| SPM (GAN) (Goodfellow et al., 2014a) | 38.7 | 32.7 |
| GAN (Goodfellow et al., 2014a) | 38.6 | 32.8 |
| SPM (LSGAN) (Mao et al., 2016) | 40.0 | 32.4 |
| LSGAN (Mao et al., 2016) | 39.9 | 32.5 |
| SPM (WGAN + GP) (Gulrajani et al., 2017) | 39.6 | 30.5 |
| WGAN + GP (Gulrajani et al., 2017) | 39.7 | 30.6 |
| SPM (EBGAN) (Zhao et al., 2017) | 40.2 | 32.3 |
| EBGAN (Zhao et al., 2017) | 41.5 | 32.5 |
| SPM (Our LDRSP) | 42.8 | 37.2 |
| LDRSP (Our) | **46.4** | **38.3** |

## 4.2 PREDICTION PERFORMANCE ON BINARY IMAGE SEGMENTATION

Compared with the multi-label classification task, the image segmentation task is more challenging due to the high-dimensional output space and the complex correlation among variables in the segmentation masks. We utilize the Weizmann horses dataset (Borenstein & Ullman, 2004) in this section. It is a commonly used dataset for binary image segmentation which consists of 328 left

oriented horse images and their corresponding binary segmentation masks. Following (Gygli et al., 2017; Li et al., 2013), all images and segmentation masks are resized to $32 \times 32$. The segmentation of horses at this low resolution is challenging and requires models to capture strong priors of the horse shape, since some thin parts of the horse like legs, tails are almost invisible in the images. We follow the experimental protocol of (Li et al., 2013) to split the Weizmann horses dataset and report results on the same testing set.

In the experiment, we adopt the fully convolutional network (FCN) (Long et al., 2015) baseline model proposed in (Gygli et al., 2017) as our structured prediction model. It consists of three $5 \times 5$ convolutional layers and two deconvolution layers. We find that using discriminators similar to PatchGAN discriminators (Isola et al., 2017; Liu et al., 2017) improves the prediction performance. Therefore, our discriminators are designed to map $(\mathbf{x}, \mathbf{y})$ to score matrices. Here $\mathbf{x} \in R^{W \times 3}$, $\mathbf{y} \in R^{W \times C}$, $W$ is the number of pixels in an image, and, $C$ is the number of object classes. We view image segmentation as pixel-level multi-label classification. Instead of approximating discriminators to the intersection over union (IOU) metric, we calculate the $F_1$ metric for each pixel and estimate an oracle value function $\mathbf{v}^* \in [0, 1]^W$,

$$v_i^* = \frac{2(\mathbf{y_i} \cap \mathbf{y_i^*})}{(\mathbf{y_i} \cap \mathbf{y_i^*}) + (\mathbf{y_i} \cup \mathbf{y_i^*})}, \tag{11}$$

where $\mathbf{y_i} \in R^C$ denotes the $i$-th row of $\mathbf{y}$. Our discriminators are implemented by modifying the FCN baseline model: the number of object classes is set to 1 and a sigmoid function is added at the end of the discriminator to ensure $v(\mathbf{x}, \mathbf{y}) \in (0, 1)^W$. Similar to the multi-label classification, a cross-entropy loss is adopted as the surrogate loss.

For both discriminators and FCNs, we adopt Adam optimizers and set the learning rates to 0.01. Data augmentation is utilized by using random $24 \times 24$ patches cropped from the $32 \times 32$ images and by randomly mirroring the images. We empirically find that setting the inference step size $\eta$ to be 4.0 and setting the number of inference steps to be 30 achieve the best performance. At the inference stage, we divide each test image into 36 crops. The proposed method is utilized to estimate segmentation masks for these crops. The estimated segmentation masks are averaged to obtain the final segmentation mask. We notice that the proposed method usually converges within 20 inference steps.

Table 2: The comparison of IOU between our LDRSP and other adversarial learning methods on the Weizmann horses dataset. FCN is adopted as our structured prediction model.

| Method | Mean IOU % | Global IOU % |
|---|---|---|
| CHOPPS (Li et al., 2013) | 69.9 | - |
| DVN (Gygli et al., 2017) | 84.1 | 84.0 |
| FCN baseline (Gygli et al., 2017) | 78.56 | 78.7 |
| FCN (GAN) (Goodfellow et al., 2014a) | 79.8 | 79.7 |
| GAN (Goodfellow et al., 2014a) | 79.7 | 79.5 |
| FCN (LSGAN) (Mao et al., 2016) | 80.3 | 79.9 |
| LSGAN (Mao et al., 2016) | 79.7 | 79.3 |
| FCN (WGAN + GP) (Gulrajani et al., 2017) | 80.6 | 80.4 |
| WGAN + GP (Gulrajani et al., 2017) | 80.5 | 80.4 |
| FCN (EBGAN) (Zhao et al., 2017) | 80.9 | 80.7 |
| EBGAN (Zhao et al., 2017) | 80.9 | 80.8 |
| FCN (Our LDRSP) | 81.3 | 81.3 |
| LDRSP (Our) | **85.5** | **85.4** |

As commonly done in the literature, we report the mean image IOU as well as the IOU over the whole testing set on the Weizmann horses dataset in Table 2. A higher IOU score means a more accurate segmentation mask. It is clear that the proposed LDRSP outperforms other state-of-the-art methods on both metrics. It can be observed that using the discriminators of the LDRSP to refine the FCN improves the performance by 4.2 % on the Mean IOU metric and 4.1 % on the Global IOU metric. It indicates that our LDRSP learns discriminators that are able to estimate stronger horse shape priors than the FCN. As it can be seen, the refinement of discriminators learned by other adversarial learning methods, *e.g.*, GAN (Goodfellow et al., 2014a), WGAN+GP (Gulrajani et al.,

2017), LSGAN (Mao et al., 2016) decreases the performance. It's due to the fact that these methods learn discriminators as classifiers which assign almost the same scores to all the predicted samples. Thus, it's difficult to utilize these discriminators to improve the performance of FCNs. Our LDRSP outperforms the DVN (Gygli et al., 2017). It indicates that utilizing the proposed method to jointly learn energy-based models and structured prediction models advances over learning energy-based models alone.

The qualitative results on the Weizmann horses dataset are shown in Figure 1. It can be seen that FCN shows poor performance in segmenting thin parts like legs and generating single-connected segmentation masks. The proposed method utilizes discriminators to refine the predictions of the FCN by filling the missing part (*e.g.*, Figure 1, seventh and eighth row, far left images), generating legs to connect disconnected parts (*e.g.*, Figure 1, seventh and eighth row, first and second images from the right).

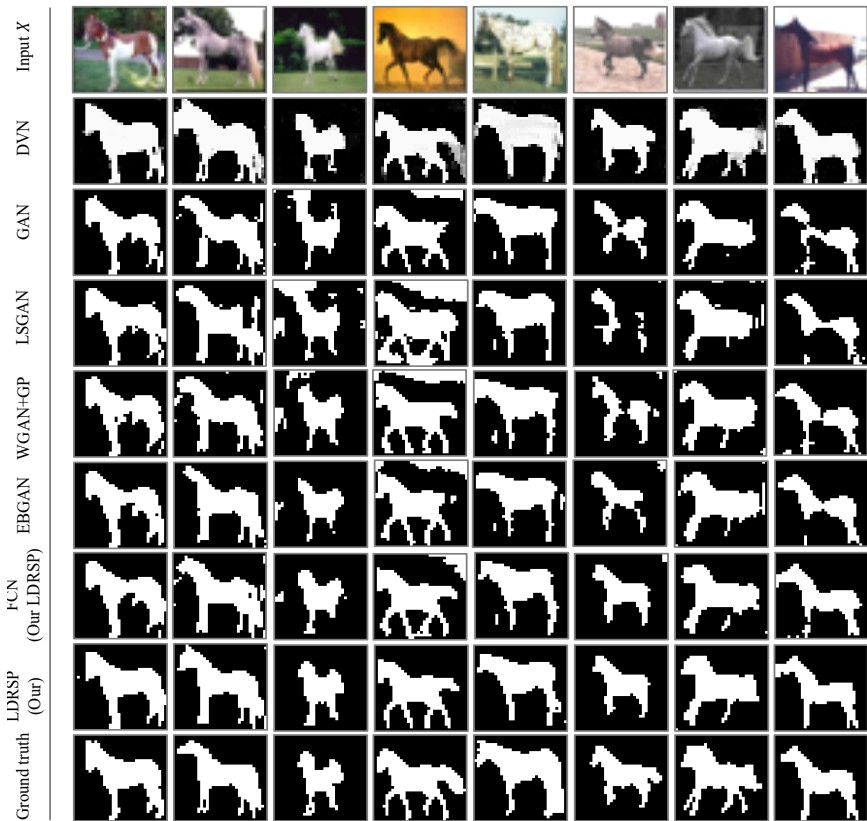

Figure 1: Qualitative results on the Weizmann $32 \times 32$ dataset.

### 4.3 PREDICTION PERFORMANCE ON 3-CLASS FACE SEGMENTATION

We utilize the Labeled Faces on the Wild (LFW) dataset (Huang et al., 2007) to evaluate our method on 3-class face segmentation. This dataset contains more than 13,000 images, which were first introduced for face recognition and later were annotated on a subset of 2,927 images for face segmentation. The annotations provide superpixel-level labels which consist of three classes: face, hair, and background. Since our method generates pixel-level labels, we map pixel-level labels to superpixel-level labels by using the most frequent labels in a superpixel as the superpixel's label following (Tsogkas et al., 2015; Gygli et al., 2017). We adopt the training, validation, and testing splits proposed in (Kae et al., 2013; Tsogkas et al., 2015; Gygli et al., 2017). The network architecture and the data augmentation are the same as those utilized on the Weizmann horses dataset.

Table 3: The comparison of superpixel accuracy (SP Acc) between our LDRSP and other adversarial learning methods on the LFW dataset.

| Method | SP Acc. % |
|---|---|
| DVN (Gygli et al., 2017) | 92.44 |
| FCN baseline (Gygli et al., 2017) | 95.36 |
| FCN (GAN) (Goodfellow et al., 2014a) | 95.53 |
| GAN (Goodfellow et al., 2014a) | 95.54 |
| FCN (LSGAN) (Mao et al., 2016) | 95.51 |
| LSGAN (Mao et al., 2016) | 95.52 |
| FCN (WGAN + GP) (Gulrajani et al., 2017) | 95.59 |
| WGAN + GP (Gulrajani et al., 2017) | 95.59 |
| FCN (EBGAN) (Zhao et al., 2017) | 95.50 |
| EBGAN (Zhao et al., 2017) | 95.52 |
| FCN (Our LDRSP) | 95.87 |
| LDRSP (Our) | **96.47** |

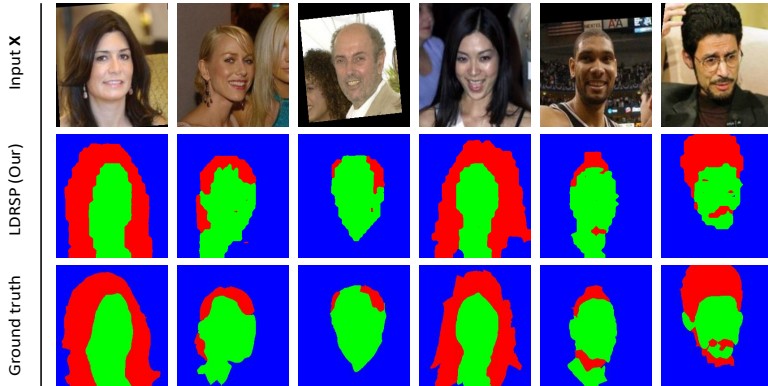

Figure 2: Qualitative results on the LFW dataset.

We compare the proposed method with other methods on the LFW dataset in Table 3. It can be observed that the FCN baseline outperforms the DVN (Gygli et al., 2017) by 2.92 % on the superpixel accuracy metric, while our method outperforms the FCN baseline by 1.11 %. The performance improvement between our method and the DVN (Gygli et al., 2017) on the LFW dataset is more significant than the performance improvement on the Weizmann horses dataset. It indicates that utilizing the proposed method to jointly learn an energy-based model and a FCN greatly improves the performance of the energy-based model, when the output space is large. The qualitative results of the LDRSP on the LFW dataset are shown in Figure 2. As it can be observed, our method can generate high-quality hair and face segmentation masks that are close to the ground-truth labels.

## 5  CONCLUSION

This paper proposes a novel learning framework, in which discriminative models are learned to refine structured prediction models. Discriminative models are trained as energy-based models to estimate scores that measure the quality of generated samples. Structured prediction models are trained to predict contrastive samples with maximum energy scores. Once models are learned, we perform inference by following the ascent gradient directions of discriminative models to refine structured prediction models. We apply the proposed method to multi-label classification and image segmentation tasks. The experimental results indicate that discriminative models learned by the proposed methods can effectively refine generative models. As the future work, we will explore different ways to generate extra training samples and apply our method to more challenging tasks.

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
