# OpenReview forum: "Learning Discriminators as Energy Networks in Adversarial Learning"
_ICLR.cc/2019/Conference_

### Official Review · AnonReviewer2 · 2018-10-30
**Interesting ideas, but needs a few more justifications for certain modeling choices over others**

**Rating:** 5
**Confidence:** 4

**Review:**

Summary:
This paper effectively learns a variant of a Deep Value Network (Gygli et al 2017), a model consisting of an energy network that assigns scores to input-output tuples that is trained to mimic a task-specific loss. The primary differences between the model presented in this work (titled LDRSP) and DVNs are twofold: first, the initial label prediction used at test time for inference is the output of a model rather than being initialized to all zeros. Second, a GAN-inspired loss is used to train both the scoring function and the initial prediction estimator. This new setup is compared against a variety of recent structured prediction methods on the tasks of multilabel classification, semantic segmentation, and 3-class face segmentation.

Comments:
I think the ideas presented in this paper are interesting, but I think their presentation could be a bit clearer. As mentioned in the summary, what you’re presenting is still more or less a deep-value network with some additions - however, you don’t refer to it as such in the body of the paper anywhere I saw. The first addition is the use of a learned model to produce the initial prediction; this is a natural extension to Deep Value Networks, and on its own is somewhat incremental in nature. I do not think you adequately explained why you chose to use a GAN-like loss to learn these models. Another baseline that would have helped justify its use would be to train your G model to predict structured outputs in the standard way (max-margin or cross-entropy loss) and then train your energy function in the DVN way.

The experimental settings are somewhat small in scope but follow the precedent set by previous structured prediction papers, which is fine. You make appropriate comparisons against previous structured prediction models as well as against different types of GAN-like losses. But, as I mentioned before, I think you needed to have more comparisons against different ways of training these networks that do not follow a GAN-inspired framework.

Overall, I like the new ideas in this paper but I think a few more experimental settings are required before they should be published.

=== after rebuttal ===

I appreciate the response, but I still think further analysis of the model is needed to understand where the gains in performance are coming from. The claim is that this is due to the adversarial loss used, but without further ablations I feel this is too strong a claim to be making given the current evidence.

---

> ### Author Response · Authors · 2018-11-26
> **The motivation of this work.**
>
> Thanks for saying that our idea is interesting.
>
> This work focuses on adversarial learning rather than structured prediction. We propose a novel adversarial learning framework and test it on structured prediction tasks. We notice that few works have applied discriminators D to refine generators G at the inference stage. This work proposes to learn D as energy-based models and our D can be used to effectively refine G. The training algorithm of D is similar to Gygli et al. (2017). However, the overall adversarial learning framework is novel.

---

### Official Review · AnonReviewer1 · 2018-11-01
**Useful but incremental improvement over previous work; Clarity of writing needs improvement**

**Rating:** 5
**Confidence:** 4

**Review:**

- The writing and structure of the paper can be improved. It is difficult to read without first reading Gygli et al. 2017, and this paper should be more self-contained. There are also many parts that are not clear:
  1. What is the model structure of G? Is it another neural network, or other structured prediction approaches such as graphical models?
  2. The GAN-based approaches listed in the experiments section are originally designed for learning generative models. What are the adaptations required to turn them into structured prediction models? This is not clear at all.

- The convergence of GAN training is an ongoing research problem and in practice also affects the quality of results. Yet in this paper I don't see any details on how these adversarial networks are trained jointly (e.g., heuristics to balance the progress on G and D). The authors should give more details on these.

- The main difference of this paper compared to Gygli et al. 2017 seems to be the joint learning of a prediction model G. Instead of relying only on the valuation network D and starts iterative gradient ascent on the initial prediction of a vector y^(0) of all zeros, the authors start the iterative gradient ascent with the prediction from G (equation 7). Otherwise the paper looks very similar to Gygli et al. 2017, including the training sample generation methods. So to me the main message of this paper is that you can improve deep value networks by providing a better starting point in inference with G. The improvement is somewhat small though, between 1-2% on the datasets shown in the experiments section.

- Overall this paper gives a useful but incremental improvement over the deep value network proposed by Gygli et al. 2017. However, the writing should be substantially improved to make the paper more self-contained and to include missing experiment details.

=== after rebuttal ===

The authors explain some of their model choices in the rebuttal, but I am still not convinced about the difference with Gygli et al. 2017 is significant enough.

---

> ### Author Response · Authors · 2018-11-26
> **The motivation of this work and missing details**
>
> Thanks for your comments.
> The structure of G: The model structure of G is a neural network. The network architecture can be found in section 4.
>
> Why GAN-based approach are listed in experiments: It has been shown that adversarial learning can be utilized to improve the performance of structured prediction models (Luc et al., 2016; Hwang et al., 2018). This work proposes a novel adversarial learning framework and tests it on structured prediction tasks. Therefore, we compare the proposed method with existing state-of-the-art adversarial learning methods in this work.
>
> Training details: For each training iteration, we update D for k steps and update G for one step. We set k to 1,2,5 in the experiments and found that setting k to different numbers slightly changed the performance. We set k to 1 in this work.
>
> The difference between this work and ( Gygli et al. 2017): This work focuses on adversarial learning rather than structured prediction. We propose a novel adversarial learning framework where D can be utilized to refine G at the inference stage. The framework is tested on structured prediction tasks. We find that D learned as energy-based models can be used to effectively refine G. The training algorithm of D is similar to Gygli et al. (2017). However, the overall adversarial learning framework is novel.

---

### Official Review · AnonReviewer3 · 2018-11-02
**Adversarial learning framework for energy-based structured prediction**

**Rating:** 5
**Confidence:** 5

**Review:**

Building on the work of (Gygli et al., 2017), this paper introduces a training algorithm for energy-based models for structured prediction. Similar to Gygli et al. (2017), they train an energy-based discriminator, which matches the energy value of structured outputs with their target values assigned by a value function.
The authors present the learning algorithm in an adversarial learning framework by describing a structured prediction model G and a discriminator D. However, it is very confusing for me to understand the proposed formulation as an adversarial framework. In an adversarial framework, at the equilibrium, G could be used as a final prediction model, however, the predicted output of G are still low quality. For example, considering Table 1, the performance of G is exactly the same as NN baseline, which suggests that only the first term of Eq. 6  participates in the training of G (because L_g(G, y*) is the exact objective of the NN baseline).
What that I can easily relate to, however, is that this training algorithm is similar to Gygli et al. (2017), but uses G to get an initial point for gradient-based inference. We want this initial point to be close to the target value Eq. 6. We use the initial value (prediction of G) and the ground truth as the matching constraints (Eq. 5) (as well as the other samples that construct Eq. 10). This actually describes why D can refine the output of G (because it looks at it as an initial point that needs refinement), but the discriminators in the other adversarial frameworks can't refine that much (since they have reached the equilibrium). I would love to hear authors comments on my concern regarding the proposed adversarial framework.

Other comments:
1) Lg is a surrogate loss, not the task-loss. Task-loss could be F1, IOU, BLEU, etc, which is the ultimate performance measure on a task.
2) The authors refer to G as a structured prediction model but starting from Section 4, they have switched to call it a classifier, which is confusing.
3) "Gygli et al. (2017) found that the key to learning energy-based models is generating proper training data.":  Is this a general statement for every energy-based model? I understand its effect when matching values, but is it still true for other training algorithms such as structural SVM training (Belanger and McCallum, 2016)? Do you have evidence to support it?
4) "In the experiment, we adopt the fully convolutional network (FCN) (Long et al., 2015) baseline model proposed in (Gygli et al., 2017) as our segmentation network. It consists of three 5 × 5 convolutional layers and two deconvolution layer.": The text from Gygli et al. (2017) says three convolutional layers and two fully connected layers. Are you using the same architecture? If not, can you describe the architecture in more details?
5) The qualitative results for Gygli et al. (2017) appear in https://gyglim.github.io/deep-value-net/. The reported output for DVN row is significantly worse than the segmentation results of the same horses specifically for columns 4 and 8, while the overall reported IOU in Table 2 is exactly the same. Can you describe the source of this disagreement?
6) Is having a continuous domain for value function v essential for the proposed training algorithm?

=== After rebuttal ===
I am not convinced that the improved performance is because of the adversarial training. I trained a simple MLP and with the right amount of regularization it gets 42.0% f1 score on Bibtex, so I am not sure that the adversarial training is very essential here.

---

> ### Author Response · Authors · 2018-11-26
> **We posted a revised version and addressed your comments.**
>
> Thanks for your comments.
> Adversarial learning: As Table 1 shows, the proposed adversarial learning framework actually greatly improves the performance of the structured prediction models G. The NN baseline achieves 38.9% F1 score on Bibtex and 33.8% F1 score on Bookmarks, while the G learned by the proposed method achieves 42.8% F1 score on Bibtex and 37.2% F1 score on Bookmarks. Although the adversarial learning can improve the performance of G, this work indicates that the discriminator D can be utilized to further improve the performance of G. After utilizing D to refine G, our model achieves 46.4% F1 score on Bibtex and 38.3% F1 score on Bookmarks.
>
> The difference between this work and (Gygli et al. 2017): This work focuses on the adversarial learning rather than the structured prediction. We propose a novel adversarial learning framework and test it on structured prediction tasks. In adversarial learning, we notice that the discriminators D can always successfully distinguish the ground-truth labels and the predicted labels. We propose to utilize D to improve the performance of G at the inference stage.  Former adversarial learning work treats D as classifiers, and their D can't refine G much due to the vanishing gradient issue. This work learns D as energy networks proposed in Gygli et al. (2017). We can effectively refine G by following the ascent gradient directions of D. The training algorithm of D is similar to Gygli et al. (2017). However, the overall adversarial learning framework is novel.
>
> Other comments:
> 1) Thanks for your advice. We changed task-specific loss to surrogate loss in the revised version
> 2) Thanks for your advice. We called G as structured prediction model in Section 4 in the revised version.
> 3) This statement is true for deep value networks. For other methods, e.g., SPEN (Belanger and McCallum, 2016), they need to find an output y_p that minimizes Equation (9) of their work. We modified this statement in the revised version.
> 4) Our segmentation network has the same architecture as the FCN baseline model proposed in Gygli et al. (2017). They consist of three convolutional layers and two deconvolutional layers. The first convolutional layer is 5x5 with stride 1, and other convolutional layers are 5x5 with stride 2. Our discriminator network also consists of three convolutional layers and two deconvolutional layers.
> 5) We reimplemented the DVN proposed in Gygli et al. (2017). Our implementation achieved 82.2% on the mean IOU metric and 81.8% on the global IOU metric. The qualitative results are generated using our code. We didn't notice that Gygli et al. (2017) had released their qualitative results. We have replaced the qualitative results in the revised version.
> 6) Yes. If the value function v has a continuous domain and discriminator D is learned to approximate v, we can follow the ascent gradient direction of the discriminator to refine the structured prediction network.

---

> > ### Public Comment · (anonymous) · 2018-12-09
> > **Detail on segmentation network**
> >
> > Thanks for providing details on the segmentation network. I was just wondering what are the number of filters and strides for the deconvolutional layers.

---

> > > ### Author Response · Authors · 2018-12-09
> > > **Details on deconvolutional layers**
> > >
> > > There are two deconvolutional layers. For the first deconvolutional layer, the stride is 2, the kernel size is 4, the number of filters is equal to the number of classes.  For the second deconvolutional layer, the stride is 4, the kernel size is 8, the number of filters is equal to the number of classes.

---

### Public Comment · ~Kevin_Gimpel1 · 2018-10-01
**you may be interested in our related paper**

Nice paper! We wanted to bring to your attention our ICLR 2018 paper (Tu and Gimpel, "Learning Approximate Inference Networks for Structured Prediction", https://arxiv.org/abs/1803.03376 ) which similarly combines structured prediction and a GAN-like learning formulation with separate networks for scoring and inference. There are some differences in the actual objectives used (we use hinge losses with a value (cost) function while you use regression to predict the value function directly if I understand correctly), and we use inference networks for amortized/distilled inference while you additionally use gradient descent on the output of your generator. But in general, glad to see more evidence that adversarial learning frameworks are effective for structured prediction!

---

### Public Comment · (anonymous) · 2018-10-24
**question about Mean-IOU and Global-IOU**

I have a question about the evaluation metrics: mean-IOU and global IOU. I cannot find the metric global IOU online, and I am not in the CV field.  Could you please give a formal definition, or tell us how do you compute it ?

In the paper, you mention that the mean-IOU is mean image IOU, and global-IOU is IOU over whole test dataset. For mean-IOU, do you mean to compute the foreground IOU for each test image, and then take the average ?  Could you please explain these two metrics more clearly? Thanks a lot.

---

> ### Author Response · Authors · 2018-10-25
> **mean IOU and global IOU**
>
> Thanks for your interest in our paper. The mean IOU is computing the foreground IOU for each test image, and then taking the average [2]. The global IOU is  |Y \cap Y^| / |Y \cup Y^|, where Y and Y^ are the sets of ground truth and predicted foreground pixels [1].
>
> [1] Yang, Jimei, Safar, Simon, and Yang, Ming-Hsuan. Maxmargin boltzmann machines for object segmentation. CVPR, 2014.
> [2] Y. Li, D. Tarlow, and R. Zemel. Exploring compositional high order pattern potentials for structured output learning. In CVPR, 2013.

---

> > ### Public Comment · (anonymous) · 2018-10-25
> > **Mean IOU**
> >
> > Thanks for your reply.
> >
> > However, I find that the mean-IOU should be defined as the average over IOUs of all classes [1]. For the case of horse experiment, should the background be counted as one class ?
> >
> > [1]. Long, Jonathan, Evan Shelhamer, and Trevor Darrell. "Fully Convolutional Networks for Semantic Segmentation."

---

> > > ### Author Response · Authors · 2018-10-25
> > > **Mean IOU**
> > >
> > > For the horse segmentation, former works [1,2] only reported the foreground IOU. To make a fair comparison, we followed their settings and only reported the foreground IOU.
> > >
> > > [1] Y. Li, D. Tarlow, and R. Zemel. Exploring compositional high order pattern potentials for structured output learning. In CVPR, 2013.
> > > [2] Yang, Jimei, Safar, Simon, and Yang, Ming-Hsuan. Maxmargin boltzmann machines for object segmentation. CVPR, 2014.

---

### Meta-Review · Area_Chair1 · 2018-12-18
**Needs rewriting**

**Confidence:** 4
**Recommendation:** Reject

**Metareview:**

All three reviewers expressed concerns about the writing of the paper. The AC thus recommends "revise and resubmit".